# A Contemporary Approach to Non-Invasive 3D Determination of Individual Masticatory Muscle Forces: A Proof of Concept

**DOI:** 10.3390/jpm12081273

**Published:** 2022-08-02

**Authors:** Bram B. J. Merema, Jelbrich J. Sieswerda, Frederik K. L. Spijkervet, Joep Kraeima, Max J. H. Witjes

**Affiliations:** Department of Oral and Maxillofacial Surgery, University Medical Center Groningen, University of Groningen, Hanzeplein 1, 9700 RB Groningen, The Netherlands; j.j.sieswerda@gmail.com (J.J.S.); f.k.l.spijkervet@umcg.nl (F.K.L.S.); j.kraeima@umcg.nl (J.K.); m.j.h.witjes@umcg.nl (M.J.H.W.)

**Keywords:** 3D-VSP, CAD/CAM, FEA, finite element analysis, masticatory, muscle force, mandible, jaw, biomechanical, intrinsic strength, patient-specific, custom, bite force, muscle delineation

## Abstract

Over the past decade, the demand for three-dimensional (3D) patient-specific (PS) modelling and simulations has increased considerably; they are now widely available and generally accepted as part of patient care. However, the patient specificity of current PS designs is often limited to this patient-matched fit and lacks individual mechanical aspects, or parameters, that conform to the specific patient’s needs in terms of biomechanical acceptance. Most biomechanical models of the mandible, e.g., finite element analyses (FEA), often used to design reconstructive implants or total joint replacement devices for the temporomandibular joint (TMJ), make use of a literature-based (mean) simplified muscular model of the masticatory muscles. A muscle’s cross-section seems proportionally related to its maximum contractile force and can be multiplied by an intrinsic strength constant, which previously has been calculated to be a constant of 37 [N/cm^2^]. Here, we propose a contemporary method to determine the patient-specific intrinsic strength value of the elevator mouth-closing muscles. The hypothesis is that patient-specific individual mandible elevator muscle forces can be approximated in a non-invasive manner. MRI muscle delineation was combined with bite force measurements and 3D-FEA to determine PS intrinsic strength values. The subject-specific intrinsic strength values were 40.6 [N/cm^2^] and 25.6 [N/cm^2^] for the 29- and 56-year-old subjects, respectively. Despite using a small cohort in this proof of concept study, we show that there is great variation between our subjects’ individual muscular intrinsic strength. This variation, together with the difference between our individual results and those presented in the literature, emphasises the value of our patient-specific muscle modelling and intrinsic strength determination protocol to ensure accurate biomechanical analyses and simulations. Furthermore, it suggests that average muscular models may only be sufficiently accurate for biomechanical analyses at a macro-scale level. A future larger cohort study will put the patient-specific intrinsic strength values in perspective.

## 1. Introduction

The demand for three-dimensional (3D) patient-specific (PS) modelling and simulations has increased considerably over the past decade and is now widely available and generally accepted as a part of patient care in oral and maxillofacial surgery. Clinicians throughout the world now make use of PS modelled oral and maxillofacial implants and prostheses, e.g., reconstruction plates for oncological surgery and temporomandibular joint (TMJ) prostheses for total joint replacements (TJR). These specifically designed devices are more accurate alternatives to conventional products [1,2] and a solution for complex cases where the shelf solutions do not suffice [3]. PS designs provide a patient-matched shape to ensure a proper fit to the bony anatomy. However, the patient specificity of current PS designs is often limited to this patient-matched fit and lacks mechanical aspects related to the individual situation, or parameters, that conform to the specific patient’s characteristics in terms of biomechanical demands. Most biomechanical models of the mandible, e.g., finite element analyses (FEA), often used to design reconstructive implants or TJR devices for the TMJ, make use of a literature-based (mean) simplified muscular model of the masticatory muscles [2,4,5,6,7]. This is due to the complexity of the masticatory muscle anatomy and the inability to directly measure separate muscle forces in vivo. Unfortunately, this directly affects the overall biomechanical model specificity for each patient, which is a limiting factor when the model is used to develop a PS implant that should address personalised optimisation. Relying on such literature-based non-PS muscular models when developing PS implants might result in the same mechanical failures as observed with conventional osteosynthesis materials, e.g., osteosynthesis plate failure, stress shielding, and, subsequently, screw loosening [8]. The morphology of the masticatory system is subject to wide anatomical variations [9]; thus, utilising an average muscular model is only valid for general purposes.

Due to practical and ethical limitations on in vivo force output measurements of single muscles, it remains challenging to approximate the true maximum acting forces of the masticatory muscles. The jaw elevator muscles, consisting of the masseter, temporalis, and medial pterygoid muscles, are predominantly inaccessible to measurement techniques, such as intramuscular electromyography (iEMG) and surface EMG (sEMG), that could approximate the acting forces. Both can be applied to record electrical stimuli in the muscles which, when combined with the resulting force output measurements, can be used to approximate a muscle’s acting force. The iEMG technique is, however, known to cause discomfort for the subject [10] due to the needle electrodes pinching the muscle. The effect of such invasive sensors on muscular behaviour is hard to fathom, mostly because of the inability to directly measure a muscle’s force in situ [11]. The sEMG technique reportedly suffers from a higher rate of crosstalk, i.e., misleading signals coming from neighbouring muscles [10,12]. Furthermore, there are many concerns regarding the sensitivity, applicability, reliability, and reproducibility of EMG measurements [10,13].

In 1846, Weber stated that the force of a muscle is related to the total cross-section of all the muscle fibres at a specified muscle length. This became known as the physiological cross-section (PCS) of a muscle [14]. It was suggested that the PCS is proportionally related to the maximum contractile force of a muscle, and thus could be multiplied by a certain constant to estimate a muscle’s force. The constant is called the intrinsic strength [P] as it represents a force per unit of PCS [N/cm^2^]. The resulting Formula (1) is used to calculate the muscle force (F_muscle_) and can be described as:***F_muscle_*****= *P* · *PCS***  [N](1)

Hitherto, many previous authors studied and suggested maximum values for the intrinsic strength of various muscle groups in order to determine the maximum separate muscle forces, but the intrinsic values varied widely [14,15,16,17,18]. Weijs and Hillen [19] reviewed the available literature on intrinsic strength and suggested a P-value of 37 [N/cm^2^], based on their experimental data. However, this value was determined from PCSs measured in cadavers combined with bite force data from a group of volunteers. The intrinsic strength calculation was carried out in 2D while assuming sagittal symmetry.

The P-value of 37 [N/cm^2^], determined by Weijs and Hillen [19], is still relevant as a general estimate for researchers who want a patient-specific model but only have the patient’s muscle cross-sectional data available [20]. Another value frequently found in maxillofacial literature is 40 [N/cm^2^] [21,22,23,24]. This value, initiated by Koolstra et al. [21] refers, however, to Weijs and Hillen’s [19] value of 37 [N/cm^2^].

The same relation was found using muscle cross-sectional area (CSA) [9,20,25]. The CSA, rather than the PCS of human masticatory muscles, is often used to estimate muscle force because it can be directly measured from computed tomographic (CT) or magnetic resonance imaging (MRI) data, and has been shown to correlate strongly with the total cross-sectional area of all fibres, as determined by means of dissection or PCS [9,25,26].

With this study, we aimed to propose a contemporary method to determine the patient-specific intrinsic strength value of the elevator muscles. The hypothesis is that patient-specific individual mandible elevator muscle forces can be approximated in a non-invasive manner by combining MRI muscle cross-section data, bite force measurements and 3D finite element analysis simulations, which can be used in patient-specific designs for reconstructive implants and (TMJ) total joint replacements.

## 2. Materials and Methods

### 2.1. 3D Muscular Model

Our volunteers underwent an MRI scan with a 3T MRI scanner (MAGNETOM Skyra 3T, Siemens, Erlangen, Germany) using a T1 weighed sequence (PETRA, FATSAT) and 1 [mm] slice thickness, according to our centre’s regular head and neck patient oncology protocol. The subjects were scanned while in a supine position and instructed to maintain dental occlusion throughout the scan. A manual 3D segmentation of the skull and mandible was subsequently performed in the Mimics 22.0 software (Materialise, Leuven, Belgium) to function as reference geometry for further muscle delineation. Using the Brainlab 2020 software (Brainlab, München, Germany), the temporalis, medial pterygoid, masseter pars profunda, and masseter pars superficialis muscles were delineated using the brush tool. The temporalis muscles’ CSAs were measured 10 mm cranially to the Frankfurt horizontal plane (FHP), in accordance with the method described by Weijs and Hillen [27].

The muscles were exported as standard tessellation language (STL) files, along with the manual segmentations of the skull and mandible. Next, the STL files were imported into the 3-Matic Medical 15.0 software (Materialise, Leuven, Belgium), where the muscles were wrapped and smoothed to obtain smooth structures. Subsequently, the muscle origins and insertions were determined as the contact area between the muscle delineations and the mandible and skull. A vector was drawn between the centres of gravity for each muscle’s origin and insertion surface, indicating the muscle’s acting direction. The maximum CSA was determined for each individual muscle by slicing the muscle along its defined acting direction in increments of 1 [mm] (Figure 1). The measured CSAs, in combination with the intrinsic strength values, were used to calculate the specific muscle forces. To model the muscle forces, it was necessary to assume that all muscles exert their maximum force along their determined force vectors simultaneously. A second assumption was that a single intrinsic strength value can be applied to all the simultaneously acting muscles within one subject.

The muscle delineations on MRI and the subsequent maximum muscle CSA measurements were independently performed by two individual observers (B.M. and J.S.). The inter-observer variability in cm^2^ CSA was calculated in IBM SPSS statistics version 23 (IBM corp., Armonk, NY, USA). The inter-observer variability was supported by the calculating the interclass correlation coefficient (ICC), whereby a value of <0.40 is poor, 0.40–0.59 is fair, 0.60–0.74 is good, and 0.75–1.00 is excellent [28]. This statistic test is an indicator for the reproducibility of our muscle delineation and CSA determination between different observers.

### 2.2. Bite Force Measurements

An experiment was designed to measure the total resulting force of all the elevator muscles. Intraoral scans were made of the subjects’ dentitions (Trios III, 3Shape, Copenhagen, Denmark). In order to measure the maximum isometric bite force, a spacer was placed in between the subjects’ central incisors to allow for a minor mouth opening of 15–20 [mm], resulting in bite sensor placement at the physiological optimum muscular length [29,30,31]. The intraoral scanning included both individual arches, both arches in natural maximum occlusion and the arches in a slightly open position with the spacer in situ. These scans were aligned with the MRI scan and, subsequently, the mandible was moved to match the lower dental scan of the opened position.

A bite force sensor was developed for this specific purpose (Figure 2), based on a FlexiForce A201 piezoresistive transducer or a force-sensitive resistor (Tekscan, Inc., South Boston, MA, USA). This 0.2 [mm] thick flexible sensor is 10 [mm] in diameter and its resistance reduces with increasing pressure. Using an Arduino Uno Rev3 microcontroller (Arduino, www.arduino.cc, accessed on 1 July 2021), data were collected and processed to read the applied normal compressive force. An apparatus was developed for accurate full-range calibration of the sensor. Calibration validation resulted in full-range linearity with a maximum error of 5%, measured from 30 to 560 N.

Splints were designed to fit the subject’s dentition in order to prevent damaging the subject’s dental elements and to distribute the bite force over multiple elements. This was performed in order to lower periodontal receptor stimulation and potential pain sensations which could influence the muscles’ recruitment, and to protect the dental elements, thereby encouraging the subject to apply their maximum voluntary bite force [32,33].

The sensors were located in the incisal/midline and the first pre-molar positions since these positions are relatively easily accessible and require only minimal mouth opening in order to fit the bite sensor. The sensor pockets were positioned parallel to the FHP, resulting in a registration of the bite force magnitude in a predefined direction at predefined locations. The sensor thickness was chosen so that a mouth opening of 15–20 [mm] could be established [29,30,31] (Figure 1). The splints were printed from PA12 polyamide powder (Oceanz, Ede, The Netherlands).

The maximum isometric voluntary bite force was registered in an experiment that included four separate exercises, each consisting of five load repetitions. Incisal bite force was registered, as well as both the bilateral and unilateral premolar bite forces. To avoid fatigue, a one-minute pause was taken between each measurement. For each of the four bite scenarios, the maximum bite force was determined from the five repetitions. These maximum values were used for further calculations.

### 2.3. Finite Element Model

A 3D finite element model was set up to first determine the resulting bite forces when calculating the muscular forces from the intrinsic strength value suggested by Weijs et al. [19]. These simulations functioned as a datum measurement. In the following simulations, the problem was inversed. The in vivo bite force measurements were now used as output objective values and each subject’s muscular model was scaled in output force to match these objective values and determine the patient-specific intrinsic strength value. These simulations were based on the principle of static equilibrium of forces and moments, which can be applied to an object at rest, as is the case with isometric bites.

To briefly summarise the two scenarios:(1)Use the subject’s muscle CSA and calculate the muscle forces with the intrinsic strength (P) value of 37 [N/cm], as suggested by Weijs et al. [19], and analyse the resulting bite forces.(2)Use the subject’s muscle CSA and matching measured bite forces and calculate the patient-specific intrinsic strength value.

Regarding all the scenarios described in Section 2.2, the reaction forces were measured at both condylar supports, indicating the analysed subject’s specific TMJ forces and bite force location(s).

### 2.4. Pre-Processing/Model Preparation

The manual 3D bone segmentations of the MRI data and the intraoral scans were combined with 3D models of the skull and mandible, including the dentition, in the 3-Matic 15.0 software (Materialise, Leuven, Belgium). A cancellous volume was assigned to the mandible by means of an internal shell function, resulting in a cortical thickness of 2 mm. To ensure the correct condylar positions in our simulations, the orientation of the mandible was matched to the slightly opened position of the mandible in the intraoral scan of the dentition with the spacer in situ. The final models were imported into Solidworks 2020 (Dassault Systèmes SolidWorks Corporation, Waltham, MA, USA) and converted into non-uniform rational basis spline (NURBS) solid parts using the Geomagic for Solidworks 2021 add-in (3D systems, Rock Hill, SC, USA). All the muscle insertion surfaces were copied and assigned a surface group on the mandible model so as to distribute the force equally over the entire insertion area. 

Condyle supports were used as indirectly fixed buffers to avoid over-fixation, but, at the same time, to limit the allowed condylar excursion in both the x- and y-direction of the model to allow for natural strain of the mandible. These fixtures were modelled as rectangular blocks with the condylar shape subtracted, leaving a 2 mm layer in between the condyles and the top surfaces [34]. The tops of these condylar fixtures were fixed in the x, y, and z directions and the analysed bite positions of the splints, i.e., incisal, left, and right premolar unilateral or premolar bilateral, were fixed only in the z-direction to match the bite force experiments. The contact set of cortical and cancellous portions of the mandible were considered to be bonded, and thus one part, while a non-penetrating contact set was implemented between the mandible and the condylar supports and splints. Loads were applied to the muscle insertion surfaces using the prior determined Fx, Fy, and Fz muscle force components (see Table 2 in Section 3).

Homogeneous linear elastic material properties were applied. The used Young’s modulus and Poisson’s ratio were E = 14.700 MPa, ν = 0.3, and E = 300 MPa, ν = 0.3 for the cortical and cancellous bones, respectively [35]. The articular disc properties of E = 44.1 MPa and ν = 0.4, as presented by Tanaka et al. [36] were used for the condylar supports, while the PA12 splints were assigned E = 1.750 MPa and ν = 0.4.

Parabolic tetrahedral solid mesh elements were used to discretise the model due to the complex anatomical shape of the mandible.

### 2.5. Subjects

Our workflow was applied to two male Caucasian subjects, 29 and 56 years old (y.o.), who had voluntarily undergone magnetic resonance imaging (MRI) scanning for prior research and were still available for further experiments. No subject selection was applied. Both subjects had complete and well-preserved dentitions with normal occlusions and no missing teeth apart from the third molars. None of them had clear signs of periodontal disease, pain in the temporomandibular joint or jaw muscles, or movement restrictions.

## 3. Results

### 3.1. Muscular Model

Both subjects’ CSAs were measured longitudinally along each muscle’s determined force vector. The largest CSAs were registered as listed in Table 1. The 29 y.o. subject’s mean CSAs for the masseter superficialis, masseter profunda, pterygoideus medialis, and temporalis muscles were 4.31 [cm^2^], 2.86 [cm^2^], 3.37 [cm^2^], and 6.92 [cm^2^], respectively, whereas the 56 y.o. subject had slightly larger CSAs of 5.47 [cm^2^], 2.77 [cm^2^], 3.98 [cm^2^], and 7.13 [cm^2^], respectively.

The mean inter-observer variation between the corresponding muscle CSAs, delineated and measured by the two observers, was 0.73 cm^2^ with an interclass correlation coefficient (two-way mixed) of 0.91, indicating an excellent match of measurements by both observers [28]. Since this study only includes measurements in two subjects, no further statistical analysis was carried out.

The direction of each muscle, as described by the vector in between the centres of gravity of the origin and insertion surfaces of each muscle, were found through the Fx, Fy, and Fz components in Table 2. The FHP functioned as the x–y plane with its positive x-axis pointing anteriorly, the positive y-axis pointing towards the left side of the mandible, and the z-axis pointing cranially. The origin of the coordinate system was set where the mid-sagittal plane coincided with the FHP.
jpm-12-01273-t002_Table 2Table 2Both subjects’ muscle force vector components for the literature-based intrinsic strength value (P = 37) and the determined patient-specific intrinsic strength values (P = 40.6 and P = 25.6).



**P = 37 [N/cm²]****P = 40.6 [N/cm²]**
**Muscle****Laterality****CSA [cm²]****∑ Force [N]****Force Components****[N]****∑ Force [N]****Force Components****[N]****Subject 1, 29 y.o.**



**x****y****z**
**x****y****z****Masseter superficialis**Right4.64171.7653.2224.07161.52188.2758.3426.39177.05Left3.97146.8926.6532.16140.83161.0129.2135.26154.37**Masseter profunda**Right3.14116.3114.7033.41110.44127.4916.1136.62121.05Left2.5795.076.1230.6789.78104.216.7133.6298.42**Pterygoideus medialis**Right3.34123.537.0457.22109.25135.407.7162.72119.75Left3.40125.7111.2861.27109.19137.8012.3767.16119.69**Temporalis**Right7.49277.18139.9455.13232.82303.83153.4060.43255.20Left6.34234.64113.2147.02200.07257.20124.1051.54219.30



**P = 37 [N/cm²]****P = 25.6 [N/cm²]****Subject 2, 56 y.o.****Masseter superficialis**Right5.17191.1557.1133.41179.33126.8337.8922.17118.99Left5.76213.3067.4449.60196.18141.5244.7532.91130.17**Masseter profunda**Right2.78102.8117.7931.5596.2268.2111.8020.9363.84Left2.77102.4814.3639.5893.4367.999.5326.2661.99**Pterygoideus medialis**Right4.02148.9140.6567.95126.1298.8026.9745.0883.68Left3.93145.2735.1260.36127.3996.3923.3040.0584.52**Temporalis**Right6.55242.4682.5853.03221.71160.8754.7935.19147.10Left7.72285.46105.9567.88256.23189.4070.3045.04170.01


### 3.2. Bite Force Experiments

A total of four different bite scenarios, each including five repetitions, were recorded for each subject. All the recordings were uneventful while the subjects bit as hard as they could. Only the incisal measurements demonstrated that the subjects experienced a certain amount of insecurity or pain with the highest measured forces. The splints showed a good fit and proved to offer comfortable dental protection while guiding the subject to bite at the exact location that was used for the matching FEA. Each recording involved five repetitions of the same bite position scenario. The highest peak bite force per bite scenario was used as the maximum true in vivo bite capacity at the four specified bite locations. 

All the bite forces are listed in Table 3. The ∑ F.Bite column in Table 3 describes the resultant bilateral bite force and is the sum of the right and left peak force in the bilateral experiment. The highest bite forces were registered in the 29 y.o. subject. The maximum incisal bite was 189 [N] while the maximum unilateral measurement was 345 [N] at the pre-molar location. This subject’s highest overall bilateral bite force out of the four measurements was recorded as 474 [N] and thus considered the true maximum voluntary bite force at the premolar location. Regarding the 56 y.o. subject, we recorded 79 [N], 248 [N], and 342 [N] as the highest incisal, unilateral premolar, and bilateral premolar bite forces, respectively. In both our subjects, the registered bilateral bite forces were approximately 1.4 times (1.37 and 1.38) higher than the maximum voluntary unilateral measurements at the same premolar position.

### 3.3. Finite Element Analyses

The first FEAs, four scenarios for both subjects, were set up with an intrinsic strength value of P = 37 [N/cm^2^] and functioned as reference analyses for the subsequent inversed determination of the true subject-specific intrinsic strength value for each subject. The reaction forces, measured orthogonally to the FHP, are mentioned in Table 3 under “In silico”, with P = 37 [N/cm^2^]. We observed that the intrinsic strength value used in these reference analyses was lower than the 29 y.o. subject’s actual PS intrinsic strength, while it was too high for the 56 y.o. subject.

The results of the bilateral pre-molar measurements were summed and we considered the ultimate true bite capacity of the subject at the pre-molar location (∑ F.Bite). These values were used to scale the total muscular system of the subject in the FEA. Once the right amount of scaling was achieved, the unilateral and incisal bite scenarios were analysed. The subject-specific P values were 40.6 [N/cm^2^] and 25.6 [N/cm^2^] for the 29- and 56-year-old subjects, respectively. All the post-scaling results, including the joint reaction forces of the TMJs, are presented in Table 3.

### 3.4. Maximum Mandibular Stress

When scaling the subjects’ muscular systems, FEA showed that the stresses occurring in the mandible changed drastically for the 56 y.o. subject. Even though the location of the maximum occurring stress did not change, the P = 37 [N/cm^2^] analysis showed an increase in maximum stress compared to the calculated subject-specific intrinsic strength analyses with P = 25.6 [N/cm^2^]. The maximum von Mises stresses were found in the unilateral right premolar scenarios and occurred at the contralateral side around the mandibular oblique line. The measured values were 63.8 [MPa] for P = 37 [N/cm^2^] compared to 42.3 [MPa] in the matching P = 25.6 [N/cm^2^] scenario. Figure 3 visualises this comparison.

## 4. Discussion

We propose a contemporary method to determine the patient-specific intrinsic strength value of the elevator muscles of the mandible. Furthermore, we show how to patient-specifically approximate the value of the individual mandible elevator muscles in a non-invasive manner by combining the MRI volumetric data, bite force measurements, and 3D finite element analysis simulations.

We derived the CSAs of the elevator muscles of the mandible through an indirect 3D slicing approach. We did, however, choose to apply the single-slice measurement approach to the temporalis muscle, as suggested by Weijs and Hillen [27]. This was due to the muscle’s complex fan shape, which makes it challenging to discriminate a single slice in space with the highest CSA. Our two subjects’ values correlate well with the CSAs found in the literature [22,26,37,38]. Our approach of determining a CSA for both the masseter superficialis and the masseter profunda separately, instead of the masseter as a single unit, resulted in a slightly larger total CSA due to the different angles at which the CSAs were measured for both muscle sections. This separation of both muscle sections is important since it results in two different insertion areas and thus different mechanical arm lengths, which have been found to have more impact than CSA variation [39]. This effect is most pronounced in the masseter muscle, so a case can be made that dividing the remaining elevator muscles would only impact the model’s accuracy marginally. Although several authors subdivided the temporal muscle into two or three sections, no clear anatomical separation could be observed between such portions, making the temporal multiple force vectors rather arbitrary in those cases [10,22]. Koolstra et al. [21], on the other hand, were successful and described a clear method on how to divide the temporal muscles into three sections. 

An observation we made was the ratio between the total in vivo bilateral and unilateral bite force measurements. In both our subjects, the registered combined bilateral forces were approximately 40% (37% and 38%) higher. Several studies support this observation [33,40,41]. The majority of the available bite force measurements describe the molar bite positions. We also ran comparative analyses to determine the maximum theoretical bite forces for our subjects’ molar positions using the muscular models with the patient-specific intrinsic strengths. The results from these analyses were corrected for unilateral bite using the aforementioned unilateral to bilateral ratio which should, by approximation, match the subjects’ bite capacities. The FEA shows maximum corrected bite forces at the second molar position of around 365 N for the 56-year-old subject and around 613 N for the 29-year-old subject. These values lie within the range of healthy adults with natural teeth [41,42]. Bakke et al. described a normal incisal bite force of 120–240 [N] [43]. Our youngest subject’s measures are within this range, whereas the measured force for the other subject appears rather low. Our subjects noted that regardless of the used splints, the incisal bite capacity was limited by a pain sensation around the teeth. According to our simulated incisal bite scenarios, based on the measured bilateral premolar bite, both our subjects should have been able to generate a higher bite force at the incisal position, as high as 271 and 371 [N] (Table 3). This suggests a biological inhibition which could be caused by signals from the receptors in the periodontal ligaments and mandible. This can inhibit muscle recruitment and thereby limit the generated bite force to prevent the anatomical structures from overloading [44]. The effect of local anaesthesia on the increase in bite force supports this thought [32,45]. We presume this has a greater effect on the incisal elements than on the (pre)molar elements due to their much smaller periodontal load-bearing surface, resulting in higher technical stress. 

The current generally accepted intrinsic strength P values for the jaw elevator muscles in the literature are 37 and 40 [N/cm^2^] [19,21]. In our study, we derived P values in a subject-specific manner from FEA simulations, i.e., 25.6 and 40.6 [N/cm^2^] for the 56- and 29-year-old subjects, respectively. Since the MRIs were performed in maximum occlusion, our CSA measurements were performed on the corresponding muscle lengths. The bite force measurements were, however, registered at the physiologically optimum muscular length. Assuming a constant muscular volume results in an over-approximation of the CSAs, thus giving an under-approximated intrinsic strength value. Weijs and Hillen [19] observed this as well in their experiments and suggested a gross correction. If one assumes constant muscular volume between occlusion and the physiologically optimum muscular length, a change in CSA can be calculated using the measured change in muscle length. Applying a correction factor of 10% and 15%, the measured mean muscle length difference between the occlusion and slightly opened mandible positions for our subjects resulted in a corrected P-value of 27.1 and 46.6 [N/cm^2^] for the 56- and 29-year-old subjects, respectively. This can be easily overcome for future cases by providing the subjects with splints that force the physiologically optimum mandibular muscular length while performing the MRI.

Even though our determined subject-specific intrinsic strength values correspond rather well with the values found in the literature, they show a broad variation between our subjects. This variation implies the necessity to determine the patient-specific capacity of the muscular system of the mandible. Our 56 y.o. subject’s mandibular stress values were 63.8 [MPa] for P = 37 [N/cm^2^] versus 42.3 [MPa] in the corresponding P = 25.6 [N/cm^2^] scenario. In this case, the P = 37 [N/cm^2^] intrinsic strength, as was suggested in the literature, would have resulted in an overestimation of the muscular forces, leading to a stress increase of 51% in the analysis. Using the model to, e.g., design a PS implant or (TMJ) prosthesis, could result in a radical overestimation, i.e., too bulky or thick designs, of the final implant. Such overestimations lead to PS implants that are much stiffer than necessary which, in turn, is likely to result in stress shielding of the surrounding bone and could subsequently lead to screw loosening due to stress shielding-induced bone resorption [8]. Our 29 y.o. subject’s corrected determined intrinsic strength is approximately 25% higher than that suggested in the literature. We simulated the reconstruction of a segmental defect in the mandible and found a comparable increase in the reconstruction plate’s maximum occurring stress. Depending on the applied alloy and the actual maximum occurring stress value in the plate, this 25% stress increase could mean a decrease in a plate’s life span of 10,000 to several million cycles [46], which would mean less than a week to several years of intensive loading [47].

We realise that following the protocol suggested by this study, as well as determining patient-specific intrinsic strength values, is time consuming and will therefore not always fit in with the scheduled treatment of a patient. Hence, future studies should aim to optimise and automate parts of the methods used in the protocol described herein. For example, the delineation of the separate muscles is rather time consuming and could be overcome by applying a (semi) auto-segmentation tool. Another suggestion would be to simplify the bite force measurements by using a commercially available tool.

The variation in determined intrinsic strength values for our subjects in the current proof of concept implies that true clinical intrinsic strength determination is complex and dependent on multiple factors instead of merely the CSA of a muscle. With the results of our small cohort, presented here, we do not suggest a new general intrinsic strength value to replace the currently accepted P = 37 and 40 [N/cm^2^] values [19,21]. We did, however, observe the deviation between these values and the values we determined in this study, as well as the variation we found between our subjects. Therefore, it appears necessary to determine the intrinsic strength in a PS manner when critical biomechanical models or simulations are performed. 

In the near future, we aim to start a study in which PS intrinsic strength determinations, as presented here, will be carried out for a large group of patients as part of the clinical evaluation. We aim to further study the spread of individual intrinsic strength values and to conclude if the intrinsic strength should indeed be calculated patient-specifically in all cases.

## 5. Conclusions

Despite using a small cohort in this proof of concept study, we show that there is great variation between our subjects’ individual muscular intrinsic strength. This variation, together with the difference between our individual results and those presented in the literature, emphasises the value of our patient-specific muscle modelling and intrinsic strength determination protocol to ensure accurate biomechanical analyses and simulations. Furthermore, it suggests that average muscular models may only be sufficiently accurate for biomechanical analyses at a macro-scale level. A future larger cohort study will put the patient-specific intrinsic strength values in perspective.

## Figures and Tables

**Figure 1 jpm-12-01273-f001:**
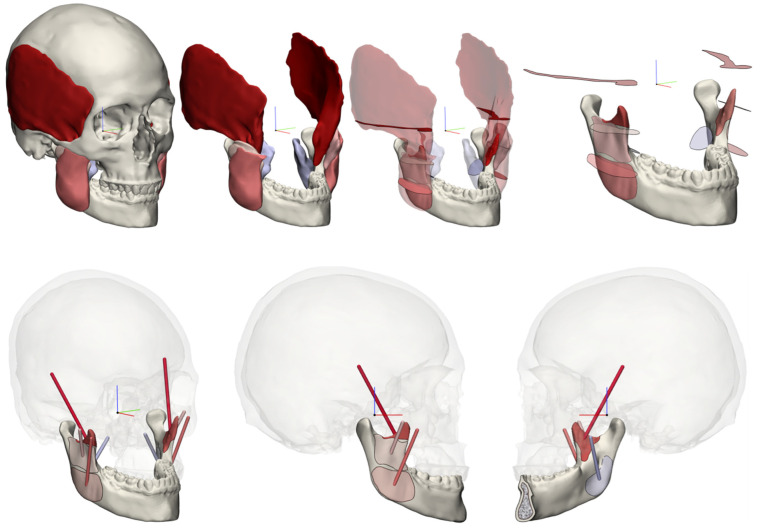
Visualisation of the smooth delineated muscles obtained from MRI data. The m. masseter superficialis, m. masseter profunda, m. pterygoideus medialis, and m. temporalis were taken into account. The muscles were sliced to determine the maximum CSA (**upper**), and the force vectors were calculated between the origin and matching insertion areas for each muscle (**lower**).

**Figure 2 jpm-12-01273-f002:**
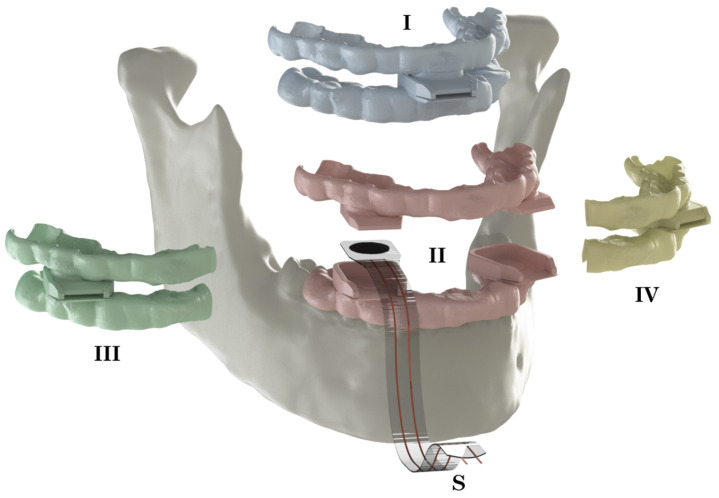
An overview of the used set-up including the bite sensor (S) and corresponding sets of upper and lower splints. The violet pair (**I**) of splints was used for incisal bite force measurements, the red pair (**II**) for bilateral premolar measurements, and the green (**III**) and yellow (**IV**) pairs for unilateral measurements of the right and left side of the premolars, respectively.

**Figure 3 jpm-12-01273-f003:**
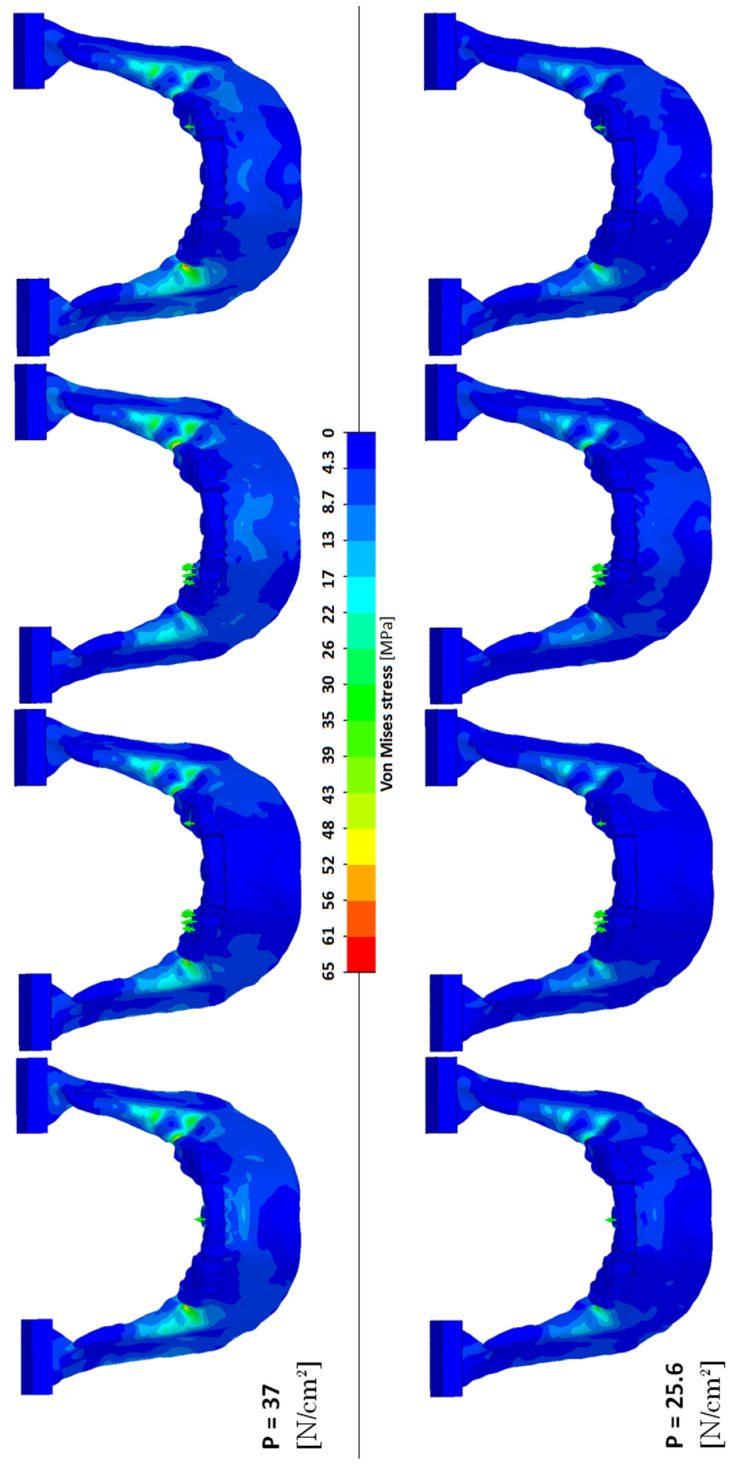
Visualisation of the von-Mises stress occurring in all the FEA scenarios of our 56 y.o. subject. Left to right: incisal, bilateral premolar, unilateral premolar right, and unilateral premolar left bite.

**Table 1 jpm-12-01273-t001:** An overview of both subjects’ measured maximum cross-sectional areas per muscle.

	**Subject 1, 29 y.o.**			**Subject 2, 56 y.o.**		
**Muscle**	**CSA [cm^2^]**					
	**Right**	**Left**	**Mean**	**Right**	**Left**	**Mean**
**Masseter superficialis**	4.64	3.97	4.31	5.17	5.76	5.47
**Masseter profunda**	3.14	2.57	2.86	2.78	2.77	2.77
**Pterygoideus medialis**	3.34	3.40	3.37	4.02	3.93	3.98
**Temporalis**	7.49	6.34	6.92	6.55	7.72	7.13

**Table 3 jpm-12-01273-t003:** Bite registrations through the bite force experiments (In vivo) and finite element analyses (In silico). All the presented forces acted orthogonally to the Frankfurt horizontal plane. The boxed values were matched to determine the PS intrinsic strength values.

**Subject 1, 29 y.o.**								
				**Premolar Laterality**			**Condyle**	
	**Bite Position**		**∑ F. Bite**	**Right**	**Left**	**Incisal**	**Right**	**Left**
	**In-vivo**							
	Bilat. premolar		**474**	256	218	-	-	-
	Premolar R		318	318	-	-	-	-
	Premolar L		345	-	345	-	-	-
	Incisal		189	-	-	189	-	-
**P = 37 [N/cm²]**	**In-silico**							
	Bilat. premolar		432	241	181	-	392	330
	Premolar R		426	426	-	-	326	402
	Premolar L		425	-	425	-	482	247
	Incisal		339	-	-	339	445	370
**P = 40.6 [N/cm²]**	Bilat. premolar		**474** (0%)	264 (+3%)	210 (−4%)	-	429	361
	Premolar R		467	467	-	-	357	440
	Premolar L		466	-	466	-	528	270
	Incisal		371	-	-	371	488	405
**Subject 2, 56 y.o.**								
				**Premolar Laterality**			**Condyle**	
	**Bite Position**		**∑ F.** **Bite**	**Right**	**Left**	**Incisal**	**Right**	**Left**
	**In-vivo**							
	Bilat. premolar		**342**	195	147	-	-	-
	Premolar R		197	197	-	-	-	-
	Premolar L		248		248	-	-	-
	Incisal		79	-	-	79	-	-
**P = 37 [N/cm²]**	**In-silico**							
	Bilat. premolar		520	257	263	-	360	416
	Premolar R		502	502	-	-	280	515
	Premolar L		510	-	510	-	453	333
	Incisal		409	-	-	409	415	473
**P = 25.6 [N/cm²]**	Bilat. premolar		**342** (0%)	168 (−14%)	174 (+18%)	-	241	276
	Premolar R		333	333	-	-	186	341
	Premolar L		338	-	338	-	301	222
	Incisal		271	-	-	271	275	314

## Data Availability

The authors declare that the data supporting the findings of this study are available within the paper.

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
