# Peer review of "A Contemporary Approach to Non-Invasive 3D Determination of Individual Masticatory Muscle Forces: A Proof of Concept"

_jpm, 2022, doi:10.3390/jpm12081273_

Round 1
Reviewer 1 Report
Dear Authors,
I’ve extensively read the manuscript titled “A contemporary approach to non-invasive 3D determination of individual masticatory muscle forces. The aim of this study was to propose a contemporary method to determine the patient-specific intrinsic strength value of the elevator muscles. The hypothesis is that patient-specific individual mandible elevator muscle forces can be approximated in a non-invasive manner by combining MRI muscle cross-section data, bite force measurements and 3D finite element analysis simulations, which can be used in pa-tient-specific designs for reconstructive implants and (TMJ) total joint replacements.
The methodology is appropriate and innovative and quite linear with recent evidences/ studies on this topic. I’ve not major concerns in this regard.
There are few aspects that should be addressed to increase the quality of the manuscript.
- A significant revision of the English syntax. In some parts the text is tough to read.
- Authors should better should better mention the potential limitation of digital assessment of bite forces, at least on the basis of the most recent literature.
- Did the authors perform any preliminary data analysis for data distribution assessment?
- Why there is not any form of statistical analysis?
- The non-invasive approach proposed by the authors is certainly one of the main strengths of the study. However, I would suggest the authors to mention recent improvement in 3D imaging technologies in assessing maxilla-facial anatomy, in particular the mandible, from CBCT scans, hereby underlying the great advantage of non-invasive approach, at least when appropriate for the diagnostic purpose.
(Lo Giudice A, Ronsivalle V, Grippaudo C, Lucchese A, Muraglie S, Lagravère MO, Isola G. One Step before 3D Printing-Evaluation of Imaging Software Accuracy for 3-Dimensional Analysis of the Mandible: A Comparative Study Using a Surface-to-Surface Matching Technique. Materials (Basel) 2020 Jun 21; 13 (12): 2798.
Leonardi R, Muraglie S, Lo Giudice A, Aboulazm KS, Nucera R. Evaluation of mandibular symmetry and morphology in adult patients with unilateral posterior crossbite: a CBCT study using a surface-to-surface matching technique. Eur J Orthod. 2020 Jan 29. pii: cjz106. doi: 10.1093/ejo/cjz106.
Antonino Lo Giudice, Vincenzo Ronsivalle, Concetto Spampinato, Rosalia Leonardi. Fully Automatic Segmentation Of The Mandible Based On Convolutional Neural Networks (CNNs) Orthod Craniofac Res . 2021 Sep 23. doi: 10.1111/ocr.12536. Online ahead of print.
Leonardi, R.; Aboulazm, K.; Lo Giudice, A.; Ronsivalle V.; D'Antò, V.; Lagravère, M.; Isola, G. Evaluation of mandibular changes after rapid maxillary expansion: a CBCT study in youngsters with unilateral posterior crossbite using a sur-face-to-surface matching technique. Clin Oral Investig 2021, 25, 4, 1775-1785.
- Authors should mention the potential impact of their approach even in pre-post treatment evaluation of specific malocclusion which can intefer with muscular function.
Lo Giudice A, Ronsivalle V, Lagravere M, Leonardi R, Martina S, Isola G. Transverse dentoalveolar response of mandibular arch after rapid maxillary expansion (RME) with tooth-borne and bone-borne appliances: A CBCT retrospective study Angle Orthod. 2020;90:680–687, doi: 10.2319/042520-353.1
F Venancio, J A Alarcon, L Lenguas, M Kassem, C Martin. Mandibular kinematic changes after unilateral cross-bite with lateral shift correction J Oral Rehabil . 2014 Oct;41(10):723-9. doi: 10.1111/joor.12199. Epub 2014 Jun 3.
- Authors should mention the potential impact of their methodology in the assessment of muscular function of subjects with different facial skeletal pattern
Lo Giudice A, Rustico L, Caprioglio A, Migliorati M, Nucera R. Evaluation of condylar cortical bone thickness in patient groups with different vertical facial dimensions using cone-beam computed tomography. Odontology 2020;108(4):669-675
Reviewer 2 Report
The study aimed to propose a method of determination the patient-specific intrinsic strength value of the mouth closing muscles. Unfortunately, the protocol is quite time-consuming and based on two subjects only. However, it is an interesting and non-invasive option. The part about the inter-observer variability is unclear. How was it calculated? Who examined both patients?
Reviewer 3 Report
First of all, I would like to thank the authors for their efforts in conducting this research. However, I would like to send you some consideration:
- The introduction begins the same as the summary. The wording of the text should not be the same in my opinion it should be modified.
- The sample of their study is very poor, only 2 patients. Moreover, he does not use any age range. There are many studies that talk about the difference in chewing strength between adults and children. Also between men and women. You should have a larger sample size to be able to study the validation of the proposed measurement.
- The discussion section is very limited since it does not compare similar studies previously performed. There are several studies on this subject and also on its clinical application, which is the most important.
- You cannot conclude with only two patients that the differences between the two are large. Nor can he validate the form of muscle measurement he has proposed.
Best Regards
Author Response
Please see the attachement.

Round 2
Reviewer 1 Report
the manuscript is suitable for publication
Reviewer 3 Report
Dear authors,
Despite the corrections made, the article is clearly a "case report". In my opinion I advise the authors to work towards a sample size in which they can demonstrate the line of research they maintain in the conclusions.
Best Regards